# The rate of epigenetic drift scales with maximum lifespan across mammals

Emily M. Bertucci-Richter [1,2] & Benjamin B. Parrott [1,2] ✉

Epigenetic drift or "disorder" increases across the mouse lifespan and is suggested to underlie epigenetic clock signals. While the role of epigenetic drift in determining maximum lifespan across species has been debated, robust tests of this hypothesis are lacking. Here, we test if epigenetic disorder at various levels of genomic resolution explains maximum lifespan across four mammal species. We show that epigenetic disorder increases with age in all species and at all levels of genomic resolution tested. The rate of disorder accumulation occurs faster in shorter lived species and corresponds to species adjusted maximum lifespan. While the density of cytosine-phosphate-guanine dinucleotides ("CpGs") is negatively associated with the rate of age-associated disorder accumulation, it does not fully explain differences across species. Our findings support the hypothesis that the rate of epigenetic drift explains maximum lifespan and provide partial support for the hypothesis that CpG density buffers against epigenetic drift.

Maximum lifespan varies broadly across the tree of life[1,2], yet the underlying mechanisms that contribute to interspecific differences in lifespan remain largely unknown. Previous work suggests the accumulation of stochastic epigenetic modifications, broadly termed "epigenetic drift", increases with age[3,4] and have been hypothesized to explain variation in maximum lifespan across species[5,6]. Epigenetic drift, which has various definitions based on the way it is quantified[7,8], is posited to result in a gradual decrease in the robustness of epigenetic patterns, a loss of epigenetic information, or an erosion of Waddington's epigenetic landscape[5]. However, the extent to which the rate of epigenetic drift, or erosion, explains maximum lifespan has not been fully explored[9,10].

In vertebrates, the addition of a methyl group onto a cytosine base typically occurs at CpG dinucleotides and functions in maintaining genomic stability and regulating transcriptional activity[11,12]. Across animal and plant species, stereotypical changes in DNA methylation can be modeled to generate predictors of both biological and chronological age[13–17]. The broad application of epigenetic clocks and the ability for cross-species assays for epigenetic age estimation suggests that epigenetic aging is likely a conserved aspect of aging across the tree of life. Even more, recent evidence demonstrates that double-stranded DNA breaks can result in the

erosion of the epigenetic landscape, acceleration of epigenetic age, and prematurely aged phenotypes in mice[18,19]. These findings support a causal role of epigenetic drift in modulating the rate of biological aging; however, the extent to which these two phenomena are linked is still being explored.

Previous work has demonstrated that the CpG density in a subset of conserved vertebrate promoters is predictive of maximum lifespan—such that increases in CpG density are associated with longer-lived species[6,20]. It has been suggested that increased CpG density buffers against stochastic changes in methylation by providing additional methylation sites to the epigenomic landscape and subsequently slowing its erosion[5,6,20]. However, analytical approaches that allow for resolving erosion of the epigenetic landscape across specific genomic regions and species are limited[7,21]. For example, previously applied measures of epigenetic discordance have revealed increases with age, but comparisons across different datasets are hampered by technical biases due to variations in sequencing read lengths[22]. Recently, novel measures of regional and sequenced methylome-wide epigenetic disorder (hereafter referred to as "global") have facilitated a less biased estimate of epigenetic drift, and are capable of being applied across different datasets[22]. Both regional and global disorders increase across the lifespan of

[1]Savannah River Ecology Laboratory, University of Georgia, Aiken, SC 29802, USA. [2]Eugene P. Odum School of Ecology, University of Georgia, Athens, GA 30602, USA. ✉e-mail: benparrott@srel.uga.edu

mice, are enriched in regulatory regions, developmental genes, and polycomb repressive complex 2 (PRC2) targets, and respond to lifespan-extending treatments, collectively suggesting that disorder is an important component of the aging phenotype in mice[22]. Interestingly, there is a significant overlap between regions that accumulate epigenetic disorder with age and those that harbor mouse epigenetic clock sites—suggesting a possible link between the two phenomena[22].

While epigenetic drift can be quantified using various metrics[3,7,8,21], here, we define epigenetic drift as a loss of epigenetic patterning as measured by concordance between proximal cytosines within individual sequencing reads – a measure we describe as "epigenetic disorder[22]". We utilize DNA methylome datasets from cohorts of known aged individuals to test the hypothesis that the rate of accumulation of epigenetic disorder explains the maximum lifespan across four mammal species with maximum lifespans ranging from 3.8 to 26.7 years. We examine age-associated epigenetic disorder dynamics across several genomic contexts including at the global, region, and gene levels, and also specifically at loci regulated by PRC2. We then investigate if species differences in CpG density underlie patterns of age-associated epigenetic disorder using comparisons across shared genes. Overall, we demonstrate that the rate of age-associated patterning of epigenetic disorder corresponds to maximum lifespan, and provide evidence supporting a role for CpG density in modifying the accumulation of epigenetic disorder across the genome.

## Results

### Epigenetic drift increases with age and maximum lifespan

RD and global disorder increase with age in rats, mice, dogs, and baboons, but the rate of accumulation differs across species. Within the rat genome, 4765 (2.72%) regions display age-associated disorder (cor ≥ |0.4|) with 3529 (2.01%) gaining RD with age and 1236 (0.70%) losing RD with age (Fig. 1a). Collectively, this represents an increase in global disorder with age in rats (lm $\beta$ = 0.0070, $R^2$ = 0.51, $F$ = 138.5, DF = 132, $p < 2.2e{-}16$; Fig. 1b). Similarly, as previously reported in Bertucci-Richter et al. 2022[22], the mouse genome has 11,819 (4.74%) age-associated regions with 10,977 (4.40%) gaining RD with age and 842 (0.34%) losing RD with age (Fig. 1c), along with a global increase in disorder with age (lm $\beta$ = 0.0060, $R^2$ = 0.33, $F$ = 76.06, DF = 151, $p$ = 4.66e−15; Fig. 1d). Of the longer-lived species, 3003 (1.48%) age-associated regions are identified in the dog genome with 1946 (0.96%) gaining RD with age and 1057 (0.52%) losing RD with age (Fig. 1e). An increase in global disorder with age is also observed (lm $\beta$ = 0.00040, $R^2$ = 0.11, $F$ = 13.59, DF = 105, $p$ = 0.00036; Fig. 1f). Similarly, the baboon genome has 155 (0.088%) age-associated regions, with 151 (0.085%) gaining RD with age and 4 (0.0023%) losing RD with age (Fig. 1g), and a global increase in disorder with age (lm $\beta$ = 0.00094, $R^2$ = 0.090, $F$ = 21.82, DF = 210, $p$ = 5.35e−06; Fig. 1h).

The rate of increase in global metrics of disorder occurs faster in short-lived species (Fig. 2a–c). Across species, disorder scales with age globally (lm $R^2$ = 0.66, $F$ = 299.1, DF = 601, $p < 2.2e{-}16$; Fig. 2a), across all covered genes (lm $R^2$ = 0.62, $F$ = 246.2, DF = 601, $p < 2.2e{-}16$;

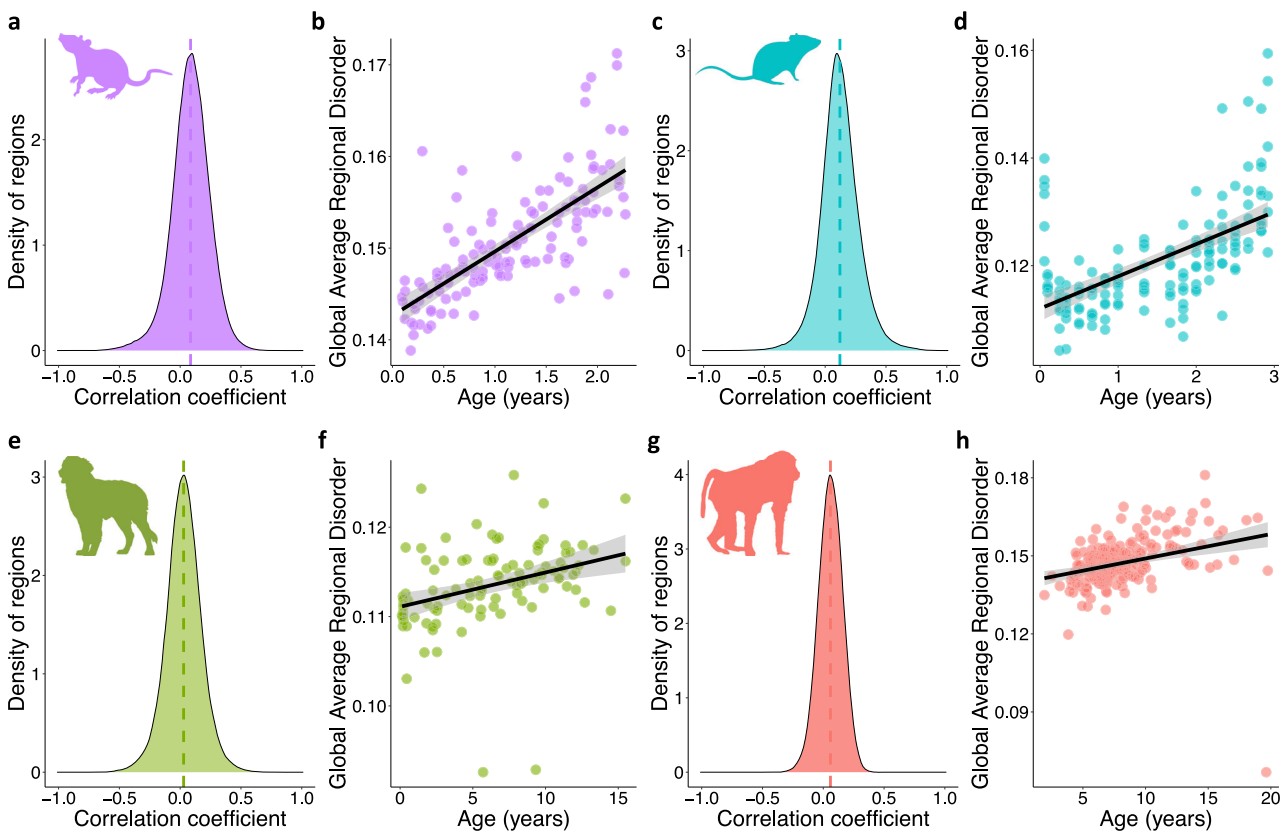

**Fig. 1 | Age-associated disorder is a shared aspect of mammalian aging.**
**a** Distribution of age-associated regional disorder across the rat genome ($n$ = 134).
**b** Global disorder of rats against chronological age in years ($n$ = 134; lm $\beta$ = 0.0070, $R^2$ = 0.51, $F$ = 138.5, DF = 132, $p < 2.2e{-}16$). **c** Distribution of age-associated regional disorder across the mouse genome ($n$ = 153). **d** Global disorder of mice against chronological age in years ($n$ = 153; lm $\beta$ = 0.0060, $R^2$ = 0.33, $F$ = 76.06, DF = 151, $p$ = 4.66e−15). **e** Distribution of age-associated regional disorder across the dog genome ($n$ = 107). **f** Global disorder of dogs against chronological age in years

($n$ = 107; lm $\beta$ = 0.00040, $R^2$ = 0.11, $F$ = 13.59, DF = 105, $p$ = 0.00036). **g** Distribution of age-associated regional disorder across the baboon genome ($n$ = 212). **h** Global disorder of baboons against chronological age in years ($n$ = 212; lm $\beta$ = 0.00094, $R^2$ = 0.090, $F$ = 21.82, DF = 210, $p$ = 5.35e−06). Species are shown by color: purple (rat), blue (mouse), green (dog), and pink (baboon). Dotted vertical lines in **a**, **c**, **e**, and **g** show the mean correlation coefficient per species. **b**, **d**, **f**, and **h** show a linear relationship with 95% confidence intervals (shaded) for age and global average regional disorder.

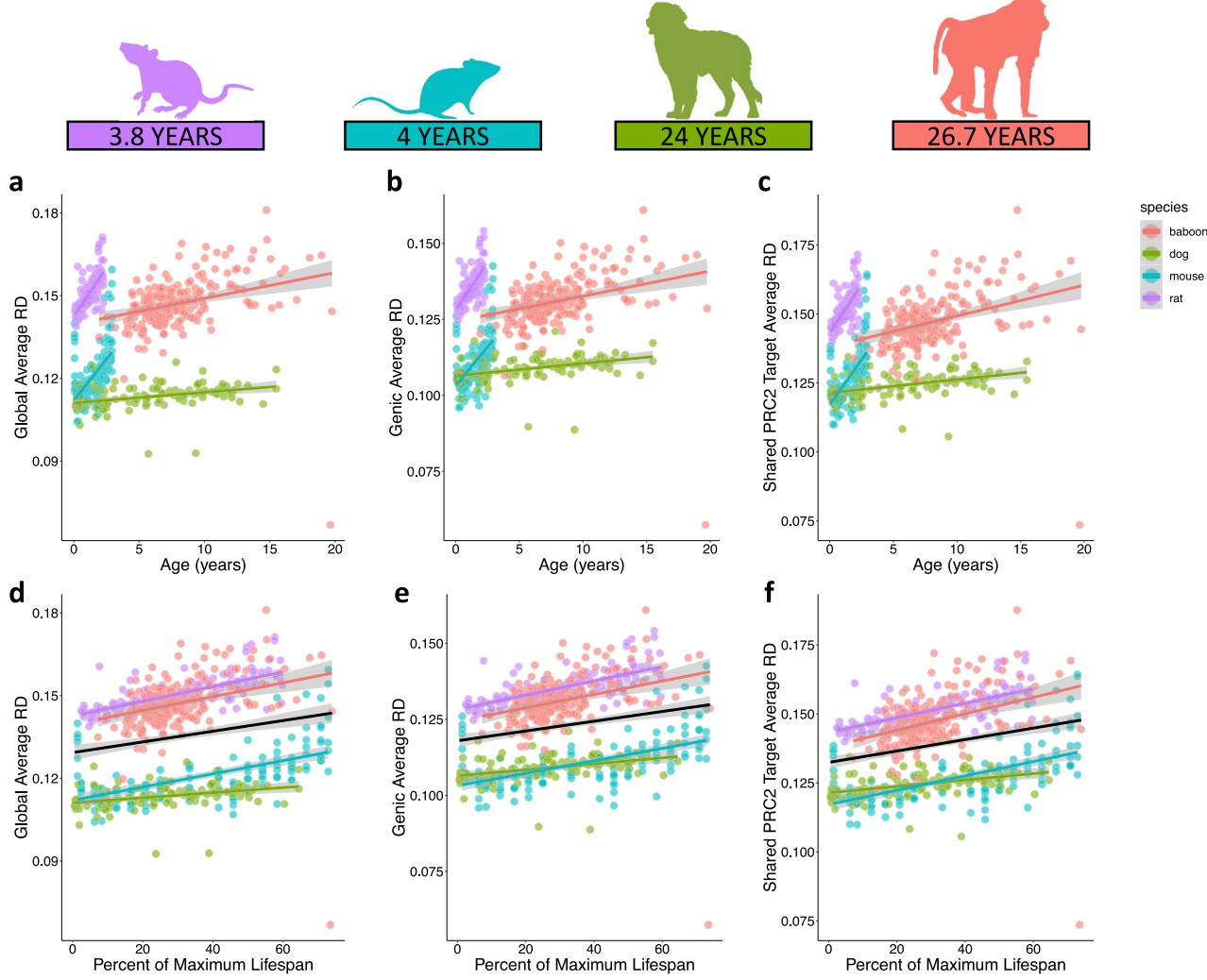

**Fig. 2 | Epigenetic disorder scales with maximum lifespan. a** Global average disorder (lm $R^2 = 0.66$, $F = 299.1$, DF = 601, $p < 2.2e{-}16$), **b** genic average disorder (lm $R^2 = 0.62$, $F = 246.2$, DF = 601, $p < 2.2e{-}16$), and **c** shared PRC2 target average disorder across chronological age (lm $R^2 = 0.60$, $F = 225.1$, DF = 601, $p < 2.2e{-}16$). **d** Global average disorder (lmm $ß = 2.21e{-}04$, SE = $1.74e{-}05$, DF = 601.1, $p < 2e{-}16$), **e** genic average disorder (lmm $ß = 1.95e{-}04$, SE = $1.54e{-}05$, DF = 601.1, $p < 2e{-}16$, and **f** shared PRC2 target (lmm $ß = 2.42e{-}04$, SE = $1.74e{-}05$, DF = 601.1, $p < 2e{-}16$) average disorder scaled across percent of maximum lifespan. Overall linear relationship is shown in black. Shaded areas represent 95% confidence intervals. Species-specific linear relationships are indicated by color: purple (rat; $n = 134$), blue (mouse; $n = 153$), green (dog; $n = 107$), and pink (baboon; $n = 212$).

Fig. 2b), and when restricted to PRC2 target genes shared across all four species (lm $R^2 = 0.60$, $F = 225.1$, DF = 601, $p < 2.2e{-}16$; Fig. 2c). Interestingly, the rate of increase corresponds to the percent of maximum lifespan accrued by an individual, regardless of species (lmm global: $ß = 2.21e{-}04$, SE = $1.74e{-}05$, DF = 601.1, $p < 2e{-}16$, Fig. 2d; lmm genic: $ß = 1.95e{-}04$, SE = $1.54e{-}05$, DF = 601.1, $p < 2e{-}16$, Fig. 2e; lmm shared PRC2 targets: $ß = 2.42e{-}04$, SE = $1.74e{-}05$, DF = 601.1, $p < 2e{-}16$, Fig. 2f).

**Epigenetic drift increases with age in non-random locations**

To test for similarities in age-associated disorder accumulation across species, we performed enrichment tests for gene ontology terms across genes included in regions with age-associated RD. We identified 2109 (rats), 3838 (mice), 1407 (dogs), and 12 (baboons) genes with age-associated RD (cor ≥ |0.4|). In rats, mice, and dogs, age-associated RD is observed in genes broadly related to DNA binding, transcription factor activity, transcriptional regulation, and developmental processes (Tables S1–S3).

There are 8440 genes represented across datasets for all four species (Fig. 3a). Within these genes, the rate of age-associated

accumulation of gene-averaged disorder (RD/year) occurs faster in the shorter-lived species (ANOVA; $F = 0.495$, DF = 3, $p < 2e{-}16$; Fig. 3b). The annual increase in RD occurs more rapidly in rats (mean rate = $0.048 \pm 0.0078$ RD/year; TukeyHSD $p < 2e{-}16$) than in mice (mean rate = $0.041 \pm 0.0059$ RD/year), and more rapidly in mice than in dogs (mean rate = $0.00034 \pm 0.0012$ RD/year; TukeyHSD $p < 2e{-}16$) and baboons (mean rate = $0.00076 \pm 0.0013$ RD/year; TukeyHSD $p < 2e{-}16$). However, RD increases more rapidly in dogs than it did baboons (TukeyHSD $p < 2e{-}7$; Fig. 3b). Two genes, EVX2 and SMAD3, contained regions with significantly age-associated RD across all four species (Fig. 3a). In datasets for each species, 10–18 regions are represented within EVX2 (rat = 10, mouse = 15, dog = 17, baboon = 18) and 15–43 regions within SMAD3 (rat = 16, mouse = 43, dog = 28, baboon = 15). Similar to gene averages, the rate of increase of RD per year across individual regions is significantly faster in shorter-lived species than in longer-lived species for regions within EVX2 (ANOVA; $F = 49.73$, DF = 3, $p = 8.4e{-}16$, Fig. 3c). However, while there is greater interspecific variation in the rate of RD accumulation in regions within SMAD3, no significant difference in the mean is observed between species (ANOVA; $F = 0.495$, DF = 3, $p = 0.69$; Fig. 3d).

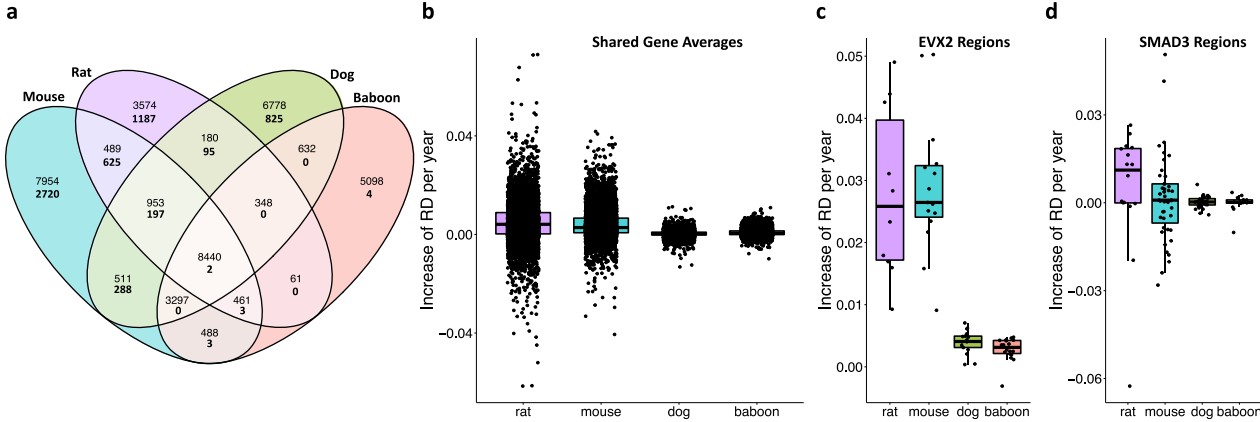

**Fig. 3 | Age-associated disorder accumulates faster in shorter-lived species.**
**a** Overlap of genes shared across species and genes with age-associated RD across species (bold). **b** Comparison of the rate of age-associated disorder (RD/year) of shared genes ($n = 8440$; ANOVA; $F = 0.495$, DF = 3, $p < 2\mathrm{e}{-16}$). Comparison across species of the rate of age-associated disorder (RD/year) across regions of **c** EVX2 ($n = 10–18$ regions within EVX2; rat = 10, mouse = 15, dog = 17, baboon = 18; ANOVA; $F = 49.73$, DF = 3, $p = 8.4\mathrm{e}{-16}$) and **d** SMAD3 ($n = 15–43$ regions within SMAD3; rat = 16, mouse = 43, dog = 28, baboon = 15; ANOVA; $F = 0.495$, DF = 3, $p = 0.69$). Species are shown by color: purple (rat), blue (mouse), green (dog), and pink (baboon). For **b**, **c**, and **d** box plots center lines represent the median, box limits represent upper and lower interquartile ranges, and whiskers represent 1.5× interquartile range with all data points plotted.

## The role of CpG density in epigenetic drift accumulation

Within the 8440 genes represented across all species, average CpG density is greater in longer-lived species (ANOVA; $F = 407.9$, DF = 3, $p < 0.002$; Fig. 4a). Mice have the lowest CpG density (mean = 0.032 $\pm$ 0.021), followed by rats (mean = 0.034 $\pm$ 0.021), baboons (mean = 0.039 $\pm$ 0.024) and dogs (mean = 0.046 $\pm$ 0.039). Mean CpG density within each gene is negatively related to rate of age-associated disorder accumulation (RD/year) according to a logarithmic relationship observed across all species (lm $\beta = -9.26\mathrm{e}{-04}$, $R^2 = 0.009$, $F = 314.8$, DF = 33758, $p < 2\mathrm{e}{-16}$; Fig. 4b). When analyzed on a per species basis, the relationship between CpG density and rate of age-associated RD is significantly negative for rat (lm $c = -0.0013$, $R^2 = 0.006$, $F = 50.24$, DF = 8438, $p = 1.47\mathrm{e}{-12}$), mouse (lm $\beta = -0.0012$, $R^2 = 0.01$, $F = 89.79$, DF = 8438, $p < 2\mathrm{e}{-16}$), and baboon (lm $\beta = -1.07\mathrm{e}{-04}$, $R^2 = 0.002$, $F = 14.81$, DF = 8438, $p = 0.00012$), and significantly positive for dog (lm $\beta = 1.28\mathrm{e}{-04}$, $R^2 = 0.005$, $F = 45.79$, DF = 8438, $p = 1.4\mathrm{e}{-11}$; Fig. 4c). We then tested whether regions which accumulate disorder more quickly in shorter lived species were the same regions wherein longer-lived species have higher CpG density. Of the 5407 genes in which disorder accumulated with age in both short- and long-lived species, 5001 (92.5%) genes accumulate age-associated disorder more rapidly in short-lived species (Fig. 4d). Of these, 3455 (63.9%) genes have greater CpG density in longer-lived species and 1546 (28.6%) genes have greater CpG density in shorter-lived species (Fig. 4d). Conversely, there are only 406 (7.5%) genes in which age-associated disorder accumulated more rapidly in long-lived species and of these, 297 (5.5%) have greater CpG density in longer-lived species and 109 (2.0%) genes have greater CpG density in shorter-lived species (Fig. 4d).

## Discussion

It is hypothesized that the rate of epigenetic drift might explain differences in maximum lifespan observed across species[5,9,10]. Using a measure of epigenetic drift, we tested this hypothesis across four mammalian species and different genomic contexts (e.g., regional and global). In addition to epigenetic disorder displaying clear relationships with chronological age, shorter-lived species accumulate epigenetic disorder more rapidly than longer-lived species. However, when chronological age is considered as a proportion of the maximum lifespan, the relationship between disorder accumulation and age is consistent regardless of species and genomic resolution. For example, epigenetic disorder accumulates within the PRC2 target, EVX2, 8.3 times faster on average in shorter-lived species when compared to

longer-lived species. Interestingly, this difference in RD rate observed in the EVX2 locus scales with differences in maximum lifespan, similar to the rates observed for global RD, mean genic RD, and PRC2 targets (Fig. 2). Thus, although DNA methylome datasets are currently only available for a limited number of taxa, it appears that the rate of RD accumulation corresponds with species-specific estimates of maximum lifespan across multiple genomic contexts. To fully assess how broadly this scaling applies, a greater representation of species across diverse taxonomic groups and maximum lifespans is needed. Collectively, the findings presented suggest progressive increases in epigenetic disorder are a robust attribute of mammalian aging and support the hypothesis that epigenetic drift is a determinate of maximum lifespan in mammals.

While all four species accumulate age-associated disorder, there are fewer age-associated regions in longer-lived species, suggesting that in addition to decreases in the rate of accumulation, preventative mechanisms might act to protect some loci in longer-lived species from epigenetic disorder. Whereas decreases in the rate of RD are likely the product of more efficient regulation of sirtuins and PRC2 complexes, CpG density, or other genomic features which are thought to slow the rate of epigenetic drift[6,18–20,23], the molecular mechanisms underlying the observed qualitative differences are more difficult to conjure. However, regardless of the number of age-associated regions, we see an enrichment of age-associated disorder accumulating in genes related to development and transcriptional regulation. This pattern closely resembles enrichments found in CpG methylation epigenetic clocks[14,15], as well as regions known to be regulated by PRC2 and sirtuin complexes[15,19,24]. Given that previous work in mice showed an enrichment of age-associated RD in PRC2 target genes[22], we also analyzed a measure of disorder in which RD was averaged across PRC2 target genes which were shared across all species. Within these shared regions, we see an increase in disorder that corresponds to both chronological age and percent of maximum lifespan, further demonstrating the conserved nature of the relationship between disorder and lifespan. We hypothesize that random processes produce epigenetic disorder but due to the relocalization of chromatin modifiers[19] and genomic architecture (e.g., CpG density[6]), disorder accumulates in non-random regions of the genome[5]. Prior studies have demonstrated that chromatin modifiers are recruited from their genomic binding sites to aid in double-strand DNA break repair[19,25] and we hypothesize that these unattended loci then disproportionately accumulate disordered DNA methylation states over time, producing the predictable

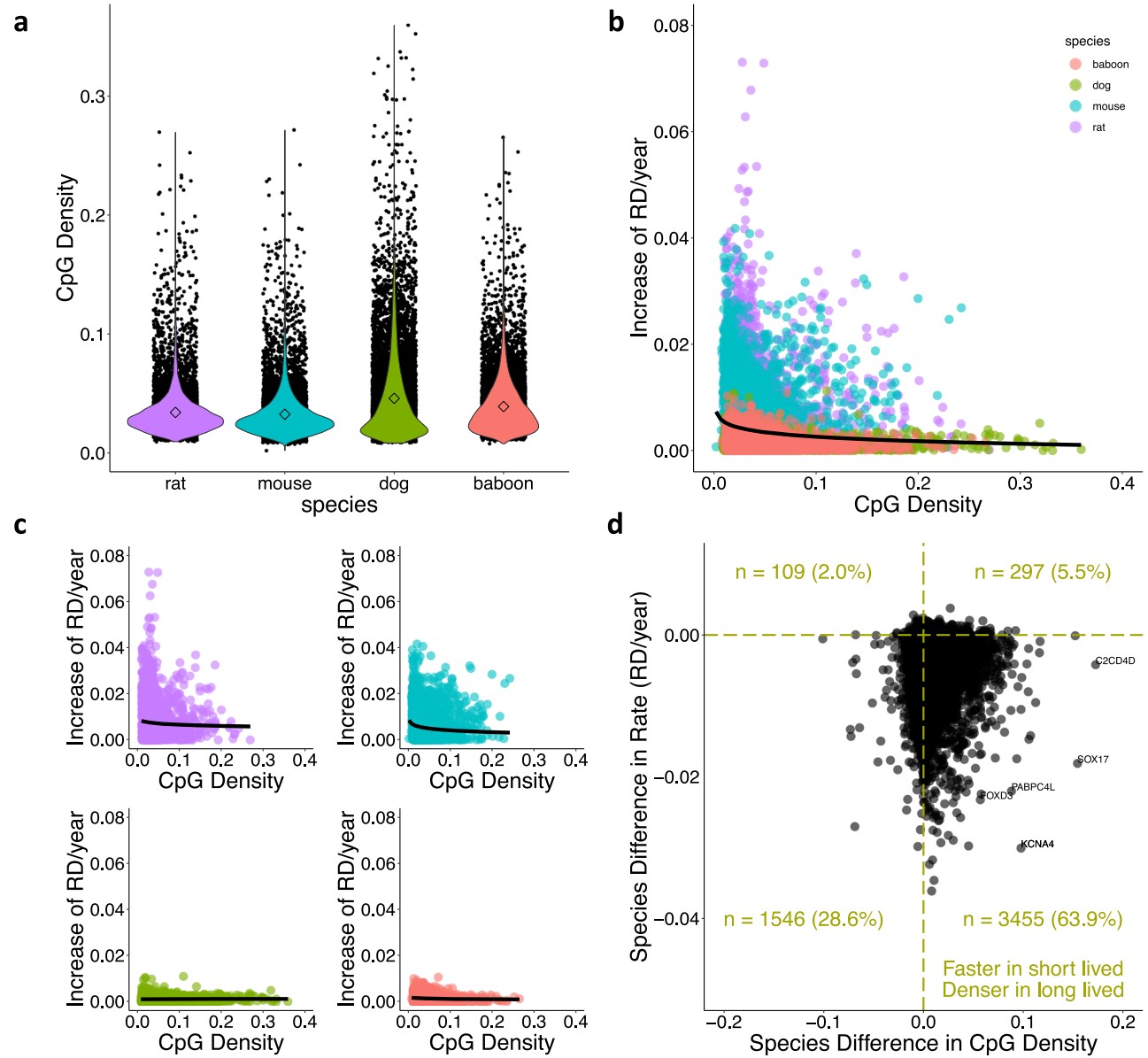

**Fig. 4 | Age-associated disorder occurs more slowly in regions with high CpG density. a** Comparison of CpG density in genes shared across species ($n = 8440$ genes; ANOVA; $F = 407.9$, DF = 3, $p < 0.002$). Diamonds show mean CpG density per species and violins represent density of data points. The relationship between CpG density and the rate of age-associated disorder accumulation (RD/year) in **b** all species (lm $\beta = -9.26e-04$, $R^2 = 0.009$, $F = 314.8$, DF = 33758, $p < 2e-16$) and **c** individual species (rat lm $\beta = -0.0013$, $R^2 = 0.006$, $F = 50.24$, DF = 8438, $p = 1.47e-12$; mouse lm $\beta = -0.0012$, $R^2 = 0.01$, $F = 89.79$, DF = 8438, $p < 2e-16$; dog lm $\beta = 1.28e-04$, $R^2 = 0.005$, $F = 45.79$, DF = 8438, $p = 1.4e-11$; baboon lm $\beta = -1.07e-04$,

$R^2 = 0.002$, $F = 14.81$, DF = 8438, $p = 0.00012$). The relationship between the rate of age-associated disorder accumulation (RD/year) and log(CpG density) is shown in black. For **b** and **c**, only genes with positive slopes (RD/year) are plotted ($n = 5407$ genes). Species are shown by color: purple (rat), blue (mouse), green (dog), and pink (baboon). **d** Classification of shared genes according to their relationship between CpG density and maximum lifespan, and rate of age-associated disorder accumulation (RD/year) and maximum lifespan. Positive values indicate that longer-lived species have higher CpG density and a faster rate of age-associated RD, respectively. The number of genes in each quadrant is listed.

patterns of disorder accumulation we demonstrate here across species.

Consistent with previous findings, we demonstrate that longer-lived species have greater CpG densities across shared genes[6,20]. Interestingly, we also find that CpG density is negatively related to the rate of disorder accumulation both across and within species, supporting the hypothesized role it plays as a buffer against the accumulation of deleterious epigenetic drift[5,19,20]. In the majority of shared genes (92.5%), the rate of disorder accumulation is greater in shorter-lived species, and of these, there are more than twice as many genes that have greater CpG density in longer-lived species. However, even at

regions of relatively low CpG density, disorder accumulates more slowly in longer-lived species, suggesting that long-lived species have additional mechanisms that protect against age-associated disorder—possibly sirtuin efficiency and DNA repair mechanisms, both of which have been previously demonstrated to correspond with mammalian lifespan[23].

Epigenetic disorder can be thought of as a form of mild damage to the epigenome—changes in the methylation status of a single CpG are likely to have little effect on phenotypes until later in life when they are cumulatively realized across a region. At that point, as would be hypothesized by the deleteriome model of aging[26], the accumulated

disorder may contribute to aging phenotypes such as transcriptional dysregulation[21,27]. Supporting this idea, previous work has demonstrated that disordered methylation in cancer cells is related to transcriptional noise[21]. Given previous work that shows the relationship between RD and epigenetic clock sites[22], along with the evidence that experimental erosion of the epigenetic landscape is sufficient to accelerate biological aging[19], our work provides further support that conserved aspects of epigenetic aging may be rooted in epigenetic drift. While we focus here on disordered patterns in DNA methylation, it would be reasonable to extend these analyses to the consistency of other epigenetic modifications that play similar roles in maintaining genomic stability. This would undoubtedly provide a broader view of epigenetic disorder and may open the possibility to analyze other species, particularly those with little to no CpG methylation, such as *Drosophila*[28]. Although more work is needed to address whether epigenetic disorder is a driver or byproduct of aging, we add to the growing empirical support that the loss of epigenetic information is a conserved aspect of mammalian aging and may be controlled by CpG density between species.

## Methods

### Data acquisition
Publicly available reduced representation bisulfite sequencing (RRBS) data were acquired from NCBI's Sequence Read Archive. All data were collected with the approval of the respective studies' institutional review boards. We utilized RRBS data from male mice (*Mus musculus*; $n = 153$; BioProject ID: PRJNA319643)[29], male rats (*Rattus norvegicus*; $n = 134$; BioProject ID: PRJNA675651)[30], male and female dogs (*Canis lupus familiaris*; $n = 107$; BioProject ID: PRJNA612432)[31], and male and female baboons (*Papio cynocephalus*; $n = 250$; BioProject ID: PRJNA648767)[32]. Sample collection and library preparation methods are detailed in refs. 29–32, respectively. Longitudinal baboon samples were removed from all analyses to prevent pseudo-replication.

### Data processing
Raw reads were trimmed of low-quality bases and sequences using Trim Galore! (v0.6.5, options for mouse and dog: --paired –rrbs –quality 25 –illumina, options for rat and baboon: –rrbs –quality 25 –illumina). Reads were then aligned either paired-end (mouse and dog) or single-end (rat and baboon) to their respective bisulfite indexed reference genomes using Bismark (v0.22.3). Reference genomes used were GRCm39 (mouse), mRatBN7.2 (rat), ROS_Cfam_1.0 (dog), and Panu3.0 (baboon). Following alignment, reads were sorted and output into text files using Samtools (v1.10).

### Measurement of regional and global disorder
Regional disorder was calculated as previously described in Bertucci-Richter et al. 2022[22] and describes the proportion of neighboring CpGs with different methylation states within a region. Briefly, we first measured the proportion of disordered neighbor pairs (PDN) on a per-read basis using a custom R script. Disordered neighbor pairs were determined to be adjacent CpGs within a read with discordant methylation statuses. We then mapped the individual reads to 200 bp windows of their respective genome, which were created using Bedtools' (v2.26.0) 'makewindows' function. Regional disorder (RD) for each 200 bp window was calculated as the average PDN of all reads for which a majority of the sequenced overlapped within each region. Regions with less than 5 reads were excluded from further analysis. Regions that were not represented in at least 80% of samples were also removed on a per-species basis. Global disorder for an individual was calculated as the average RD of all regions represented in at least 80% of all samples for their respective species.

### Age-associated disorder
Using the RD measurements for each 200 bp window, we ran individual Spearman correlations with chronological age on a per-species basis. To do this, we used the 'corr.test' function within the R package psych with the option to adjust significance values using the false discovery rate (FDR). The relationship between global disorder and age was performed using a linear model (lm) from the package stats in R with an interaction between age and species. Regions with cor ≥ |0.4| were considered "age-associated." While we hypothesized that the majority of the genome will accumulate disorder with age[5], we chose the stringent threshold (cor ≥ |0.4|) to maintain a focus on regions with the strongest relationships between age and disorder.

### Relationship of disorder with maximum lifespan
Maximum lifespan for mouse, rat, and dog was determined using the AnAge database[2]. Reported maximum lifespans were 4 years for mice, 3.8 years for rats, and 24 years for dogs. The AnAge database did not have maximum lifespan estimates for baboons so in place we used the 26.7-year maximum lifespan for the species reported in Tung et al. 2016[33]. Individual ages were calculated as a percentage of the species-specific maximum lifespan. For analyses using percent of maximum lifespan, we used a linear mixed effects model (lmm) with species as a random effect to allow for differences in intercept across species using the R packages lme4 and lmerTest. For comparable cross-species analyses, we then annotated each of the regions using their respective UCSC gene tables and further limited the analysis to include only regions that resided within genes and only regions that resided within genes considered to be PRC2 target genes in mice (as previously annotated[34]) and which were shared across all four species. Of the shared genes ($n = 8440$), we calculated the average RD for each gene and each individual. We then used linear models to determine the relationship between each gene's average disorder and age in years. The rate of change in disorder per year (beta value) was extracted and compared across species using a one-way ANOVA and Tukey HSD post hoc tests.

### CpG density
We calculated CpG density (number of CpGs/total gene length) using the package seqtk_comp in Galaxy (v 1.3.1). For rats, mice, and dogs we used the smallest start coordinate and largest end coordinate for each gene in order to be inclusive of all transcript variants. For baboon, which had a greater number of unmapped scaffolds, we averaged CpG density across all transcript variants of a gene. We compared the CpG density of all shared genes across species using a one-way ANOVA and used a two-tailed *t*-test to determine the direction of the relationship between CpG density and maximum lifespan at each gene. Due to the low sample size ($n = 4$ species), we grouped rats and mice ("short-lived") and dogs and baboons ("long-lived"). We restricted these analyses to only genes that accumulate disorder with age (beta > 0) across both short- and long-lived species ($n = 5407$ genes). We repeated the same methodology to determine the relationship between the rate of age-associated RD accumulation (RD/year) and maximum lifespan.

### Genomic enrichments
We extracted the genes represented in regions with age-associated RD (cor ≥ |0.4|) for each species and tested for genomic enrichments using g:Profiler. We used a custom background for each species using all the genes that were represented in the respective datasets. For the baboon, we used the olive baboon (*Papio anubis*) as the background as it was the only available baboon species in the web tool. We then tested for overlap of genes with age-associated RD across species ($n = 2$) and isolated the regions within those genes across all species. We modeled each region against chronological age in years for each species using a linear model and extracted the slope of each relationship. The slopes,

or rate of increase in RD per year, for each gene were compared across species using a one-way ANOVA.

## Reporting summary

Further information on research design is available in the Nature Portfolio Reporting Summary linked to this article.

## Data availability

The data that support the findings of this study are openly available in NCBI's Sequence Read Archive under BioProject IDs: PRJNA319643, PRJNA675651, PRJNA612432, and PRJNA648767.

## Code availability

Custom R scripts used for regional disorder measurement are available at: https://github.com/embertucci/epigenetic-disorder.

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

## Acknowledgements

The authors would like to thank the members of the Parrott lab and Drs. Vanessa Ezenwa, Robert Pazdro, and Olin E. Rhodes for helpful comments on this project. This material is based upon work supported by the Department of Energy Office of Environmental Management under Award Number DE-EM0005228 to the University of Georgia Research Foundation. In addition, this work was supported by awards from the National Science Foundation (Award No. 2026210 to B.B.P.) and the National Institutes of Health (Award No. 1R56AG078336-01 to B.B.P.). This report was prepared as an account of work sponsored by an agency of the United States Government. Neither the United States Government nor any agency thereof, nor any of their employees, makes any warranty, express or implied, or assumes any legal liability or responsibility for the accuracy, and completeness. Or usefulness of any information, apparatus, product, or process disclosed, or represents that its use would not infringe privately owned rights. Reference herein to any specific commercial product, process, or service by trade name, trademark, manufacturer, or otherwise does not necessarily constitute or imply its endorsement, recommendation, or favoring by the United States.

## Author contributions

E.M.B.-R. and B.B.P. conceived and designed the study. E.M.B.-R. performed the analyses. E.M.B.-R. and B.B.P. wrote the manuscript.

## Competing interests

The authors declare no competing interests.
