## [Peer Review File · Nature Communications]

The rate of epigenetic drift scales with maximum lifespan
across mammalsEditorial Note: This manuscript has been previously reviewed at another journal that is not operating a transparent peer review scheme. This document only contains reviewer comments and rebuttal letters for versions considered at *Nature Communications*.

REVIEWERS' COMMENTS

Reviewer #1 (Remarks to the Author):

The authors satisfactorily addressed all my concerns in the revision.

Reviewer #2 (Remarks to the Author):

I am understanding and sympathetic towards the authors regarding the difficulties in finding appropriate data to answer the biological questions at hand. However, four species are not sufficient to make any claim about scaling laws. Additionally, while the age range is wide, the two long and short-lived species have very similar lifespans. In this context, intermediate lifespans would be more beneficial in validating the results. Mice and rats are also closely related, adding to this issue.

While the authors provided insightful discussion in response to the reviewers, only minor edits have been made to the manuscript. Please, add this discussion to the manuscript. Particularly, how a “chaotic” driver can lead to “orderly” effects is a topic that is often misunderstood. The authors’ hypothesis: “the process itself which produces epigenetic disorder is random, but due to ... disorder accumulates in non-random regions of the genome” is a reasonable one and more discussion would improve the manuscript, even if speculative.

We have responded to the reviewer comments below (responses are in blue).

Reviewer #1 (Remarks to the Author):

The authors satisfactorily addressed all my concerns in the revision.

We would like to thank this reviewer for the time they spent reviewing the earlier versions of this submission. Their suggestions have greatly improved our manuscript.

Reviewer #2 (Remarks to the Author):

I am understanding and sympathetic towards the authors regarding the difficulties in finding appropriate data to answer the biological questions at hand. However, four species are not sufficient to make any claim about scaling laws. Additionally, while the age range is wide, the two long and short-lived species have very similar lifespans. In this context, intermediate lifespans would be more beneficial in validating the results. Mice and rats are also closely related, adding to this issue.

We agree with this point and have added a qualifying statement to the discussion which outlines this shortcoming (lines 290-292).

While the authors provided insightful discussion in response to the reviewers, only minor edits have been made to the manuscript. Please, add this discussion to the manuscript. Particularly, how a “chaotic” driver can lead to “orderly” effects is a topic that is often misunderstood. The authors’ hypothesis: “the process itself which produces epigenetic disorder is random, but due to ... disorder accumulates in non-random regions of the genome” is a reasonable one and more discussion would improve the manuscript, even if speculative.

Thank you for this suggestion. We have added details about our hypothesis into the discussion of the manuscript (lines 312-319).